

# Real-time reverse transcription loop-mediated isothermal amplification for rapid detection of SARS-CoV-2

Yee Ling Lau[1], Ilyiana Ismail[2], Nur Izati Mustapa[2], Meng Yee Lai[1], Tuan Suhaila Tuan Soh[2], Afifah Hassan[2], Kalaiarasu M. Peariasamy[3], Yee Leng Lee[3], Yoong Min Chong[4], I-Ching Sam[4] and Pik Pin Goh[5]

[1] Department of Parasitology, Faculty of Medicine, University of Malaya, Kuala Lumpur, Malaysia
[2] Department of Pathology, Hospital Sungai Buloh, Selangor, Malaysia
[3] Clinical Research Centre, Hospital Sungai Buloh, Selangor, Malaysia
[4] Department of Medical Microbiology, Faculty of Medicine, University of Malaya, Kuala Lumpur, Malaysia
[5] Institute for Clinical Research (ICR), National Institutes of Health (NIH), Ministry of Health Malaysia, Putrajaya, Malaysia

## ABSTRACT

**Background:** Highly sensitive real-time reverse transcription polymerase chain reaction (RT-qPCR) methods have been developed for the detection of SARS-CoV-2. However, they are costly. Loop-mediated isothermal amplification (LAMP) assay has emerged as a novel alternative isothermal amplification method for the detection of nucleic acid.

**Methods:** A rapid, sensitive and specific real-time reverse transcription LAMP (RT-LAMP) assay was developed for SARS-CoV-2 detection.

**Results:** This assay detected one copy/reaction of SARS-CoV-2 RNA in 30 min. Both the clinical sensitivity and specificity of this assay were 100%. The RT-LAMP showed comparable performance with RT-qPCR. Combining simplicity and cost-effectiveness, this assay is therefore recommended for use in resource resource-limited settings.

## INTRODUCTION

A novel coronavirus, SARS-CoV-2, was recently identified causing pneumonia in humans, termed coronavirus disease 2019 (COVID-19). Cases of this new infection were first reported in Wuhan, China in December 2019, and the outbreak spread to more than 155 other countries in a short time (*World Health Organization, 2020a*). The World Health Organization (WHO) declared the COVID-19 outbreak as a pandemic on 11 March 2020. Several real-time reverse transcription polymerase chain reaction (RT-qPCR) methods have been developed and recommended by Centers for Disease Control of the United States (*Centers for Disease Control, 2020*) and WHO (*World Health Organization, 2020b*; *Corman et al., 2020*) for detection of SARS-like coronaviruses and specific detection of SARS-CoV-2. These methods are highly sensitive and specific but

Corresponding author
Yee Ling Lau,
lauyeeling@um.edu.my

are too expensive to be widely used in many developing countries. RT-qPCR also requires experienced personnel, maintenance of reagents in cold storage facility, and use of a high-precision thermal cycler. Loop-mediated isothermal amplification (LAMP) has emerged as a novel alternative isothermal amplification method for the detection of nucleic acid (*Notomi et al., 2000*). The LAMP assay was reported to take less than 1 h to perform at a constant temperature. Since LAMP does not require any major equipment and is simple to perform, it represents an ideal diagnostic tool for use in areas with limited resources. LAMP assays have been described for the detection of various other infectious agents including dengue viruses (*Lau et al., 2015*; *Wong et al., 2017*) and malaria parasites (*Lau et al., 2016*). In order to further reduce costs and enable detection by the naked eye, we used hydroxynaphthol blue (HNB) dye for the colorimetric detection of the amplification reaction. To our knowledge, this is the first report of the detection of SARS-CoV-2 by real-time reverse transcription LAMP (RT-LAMP) assay with HNB.

## MATERIALS AND METHODS

### RT-LAMP assay

Primers were designed using the Primer-Explorer V4 software (Eiken Chemical Co., Ltd., Tokyo, Japan) based on SARS-CoV-2 nucleoprotein (GenBank accession no MN988713.1, LC528233.1 and MT123293.1). The sequences of the selected primers were conserved among the SARS-CoV-2 sequences (Fig. S1) that do not share homology with other SARS-like coronaviruses and Middle East respiratory syndrome coronavirus sequences (Fig. S2). Reverse transcription LAMP (RT-LAMP) was carried out using Loopamp RNA amplification kit (Eiken Chemical Co., Ltd., Tokyo, Japan). The amplification was carried out in a Loopamp real-time turbidimeter (LA-320; Teramecs, Co., Ltd., Tochigi, Japan) at 65 °C for 30 min with 2× reaction mixture, five μL RNA template and species-specific primers. The primer sequences are listed in Table S1. Endpoint assessment was done by visual inspection following the addition of one μl HNB (Sigma, St. Louis, MO, USA) in the master mix; a positive amplification was indicated by a color change from violet to sky blue (Fig. 1).

### Analytical sensitivity and specificity

To determine the analytical sensitivity of the SARS-CoV-2 RT-LAMP assay, in vitro transcript RNAs were prepared using a previously published method (*Zhang et al., 2017*). Briefly, target gene fragments were cloned to a pGEM-T vector as described in the manufacturer's protocol (Promega, Madison, WI, USA). Following digestion with SalI restriction endonuclease (NEB, Ipswich, MA, USA), the plasmids were purified. In vitro transcribed RNAs were prepared with RiboMAX™ System (Promega, Ipswich, MA, USA) and digested by deoxyribonuclease (DNase) I as described in the manufacturer's protocol. The product was then purified by phenol–chloroform extraction method. Finally, in vitro transcribed RNAs were quantified by UV spectrophotometry. The limit of detection (LOD) was determined using 10-fold serially diluted in vitro transcript RNA with known numbers of nucleic acid copies (10 cp/μL, 5 cp/μL, 2 cp/μL, 1

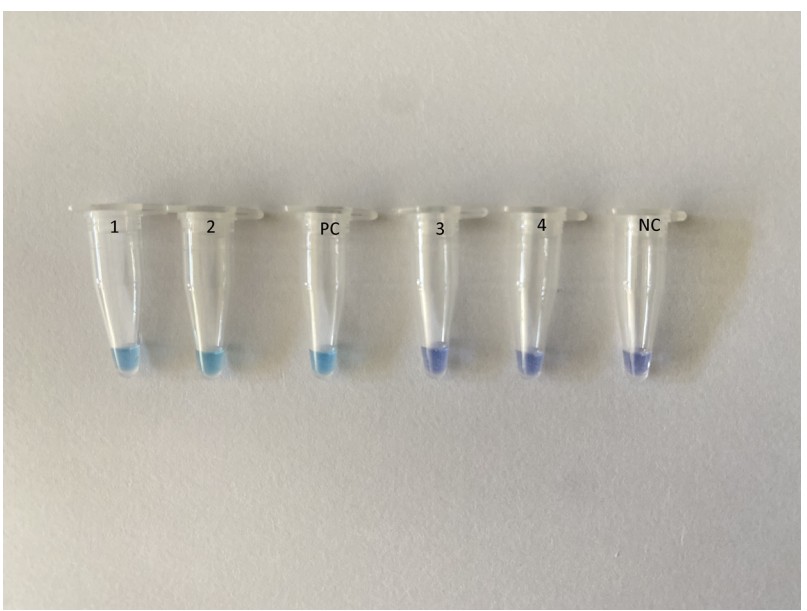

**Figure 1 Assessment of RT-LAMP results based on hydroxynaphthol blue visualization of color change.** A positive reaction indicated by sky blue color is seen in tubes 1–2 and the positive control (PC), while a violet color indicates a negative reaction in tubes 3–4 and the non-template negative control (NC).

cp/μL and 0.1 cp/μL) and comparing the assay with RT-qPCR. The reactions were carried out in duplicates. One μL of each diluted in vitro transcript RNA was used for RT-qPCR and RT-LAMP assay.

The specificity of the RT-LAMP assay was determined by using genomic RNA of coronaviruses (HCoV-OC43 and SARS-CoV), adenovirus, human metapneumovirus, influenza A (A/H1pdm2009 and A/H3) viruses, influenza B virus, parainfluenza virus 3, rhinovirus A, respiratory syncytial virus B and enterovirus D68.

## Clinical sensitivity and specificity

A total of 47 RT-qPCR positive and 42 RT-qPCR negative nasopharyngeal swabs samples were randomly chosen, regardless of RT-qPCR threshold cycle value (Ct-value) from a recent COVID-19 outbreak in Malaysia (2020) which were collected by Hospital Sungai Buloh, Malaysia. Nasopharyngeal swabs were collected using Dacron swabs made of polyester fiber. The swab was inserted into the nostril and back to the nasopharynx and left in place for a few seconds then slowly withdrawn with a rotating motion. The swab was immediately placed into a sterile vial containing two ml of viral transport media. Total RNA was extracted using QIAamp viral RNA Mini kit (Qiagen, Hilden, Germany) according to the manufacturer's instructions. In brief, 140 μL of cell lysates were transferred into 1.5 mL tubes containing 560 μL of Buffer AVL followed by vortex for 15 s. After standing at room temperature for 10 min, the collection tube was briefly centrifuged. Then 560 μL of ethanol (96–100%) was added to the sample, and mixed by vortex for 15 s. The mixture was then transferred to QIAamp Mini column and

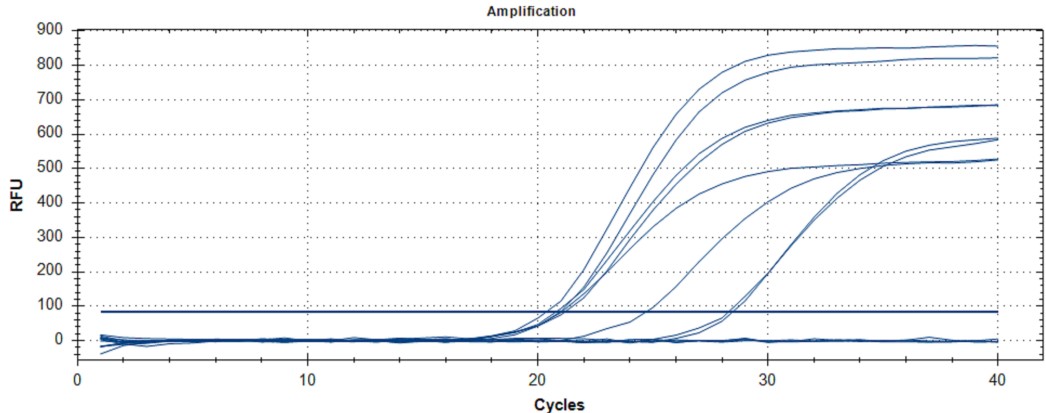

**Figure 2** **RT-qPCR results for SARS-CoV-2 detection.** The chart was generated by plotting relative fluorescence (RFU) vs. cycle number, with each colored line representing one sample. Seven positive samples and a positive control (PC) are shown with cycle threshold levels between 21 and 28 cycles.

washed with Buffer AW1 and AW2. A 50 μL-elution was obtained for each sample. The RNA samples were analyzed by RT-qPCR (Fig. 2), as previously described (*World Health Organization, 2020b*; *Corman et al., 2020*). Briefly, five μL of extracted RNA were added as template into 20 μL of the reaction mixture containing 12.5 μL of 2× reaction buffer provided with the Superscript III one step RT-PCR system with Platinum Taq Polymerase (Invitrogen, Carlsbad, CA, USA), one μL of reverse transcriptase/Taq mixture from the kit, 0.4 μL of a 50 mM magnesium sulfate solution and 1 μg of nonacetylated bovine serum albumin. Primer and probe sequences targeted at RdRP and E genes were used. Thermal cycling was performed at 55 °C for 10 min for reverse transcription, followed by 95 °C for 3 min and then 45 cycles of 95 °C for 15 s, 58 °C for 30 s.

Extracted RNA for RT-qPCR were kept in −80 °C until further analysis by RT-LAMP. SARS-CoV-2 RT-LAMP reactions were run at 65 °C for 30 min. Clinical sensitivity was calculated as (number of true positives)/(number of true positives + number of false negatives) and clinical specificity was calculated as (number of true negatives)/(number of true negatives + number of false positives) comparing to RT-qPCR. Ethical approval for this study was obtained from Medical Research Ethics Committee (MREC) Ministry of Health Malaysia (NMRR-20-535-53855).

# RESULTS

The SARS-CoV-2 RT-LAMP assay was able to detect one copy per reaction of SARS-CoV-2 RNA in 30 min while the LOD for RT-qPCR was five copies per reaction (Table S2). The time taken for amplification did not change with or without the addition of HNB in the master mix (Table S2). No cross-reactivity with other respiratory viruses (coronaviruses, adenovirus, human metapneumovirus, influenza A viruses, influenza B virus, parainfluenza virus 3, rhinovirus A, respiratory syncytial virus B and enterovirus D68) was found in either assay (Table S2).

The RT-LAMP assay demonstrated a 100% sensitivity as all the 47 RNA samples that were positive by RT-qPCR were tested positive with RT-LAMP. None of the 42 RT-qPCR negative samples were positive for SARS-CoV-2 using this assay (Fig. S3). No false–positive reactions were noted (Table S2; Fig. S4).

## DISCUSSION

The COVID-19 RT-LAMP reaction was sensitive enough to detect one copy of RNA per reaction, 5-fold better than real-time PCR. Addition of HNB to the LAMP reaction solution did not affect the time taken for amplification. Several studies have found that LAMP out-performs RT-qPCR for other viral infections (*Lau et al., 2015*; *Hu et al., 2015*; *Zhao & Feng, 2019*), which is consistent with our results.

The analysis showed that the RT-LAMP developed is 100% specific and sensitive for the detection of SARS-CoV-2 with no false positives detected. The specificity and sensitivity levels of RT-LAMP are comparable to real-time RT-qPCR methods as reported in other studies (*Reddy et al., 2012*; *Kwallah et al., 2013*). Encouragingly, the time required for confirmation of results by the RT-LAMP assay was less than 30 min, 2.5-fold faster than the time required by real-time PCR. Even including the RNA isolation step, RT-LAMP assay can be completed in less than 1 h, which is shorter than RT-qPCR (1.5 hours).

Due to its high sensitivity, RT-LAMP is prone to aerosol contamination. LAMP assays can be analyzed by running an agarose gel or adding SYBR Green I. As it has an inhibitory effect, SYBR Green I has to be added after completion of the LAMP reaction. Tubes used for RT-LAMP reactions have to be opened for gel electrophoresis for addition of SYBR Green I, which can contaminate the air and subsequent reactions. Therefore, in our study, to avoid contamination, HNB dye was used to enable interpretation of the results by the naked eye, without requiring the tubes to be opened. Addition of HNB to the LAMP reaction solution did not affect amplification efficiency. This approach has been shown to be sensitive and simple for visual detection of turkey coronavirus RNA in tissues and feces (*Cardoso et al., 2010*).

## CONCLUSION

There are several limitations in this study. First, the LOD of the SARS-CoV-2 RT-LAMP assay was not determined using serial dilutions of purified cell culture supernatant of SARS-CoV-2 due to the unavailability of a BSL3 laboratory. Secondly, although we showed specificity when testing against the most closely-related human coronavirus SARS-CoV, we were unable to obtain RNA from other coronaviruses HCoV-229E, HCoV-NL63 and MERS-CoV for specificity analysis. These experiments should be included in the future to enhance the stringency of the RT-LAMP assay.

In order to improve the efficiency of the RT-LAMP reaction, parameters such as heating temperature, dNTP concentration, and reaction time can be further optimized. There is a high risk of contamination due to the large amount of LAMP products, which often leads to false–positive results. A typical molecular laboratory should be divided into at least three areas for sample preparation, master mix preparation and product detection. However, this may not be available in resource-limited countries. To avoid contamination,

general cleaning practices such as decontaminating all surfaces with 10–15% sodium hypochlorite solution, followed by 70% ethanol should be routinely carried out. It is highly recommended to adopt a closed end-point detection method in order to avoid carry-over contamination. In addition, mineral oil inside the reaction tubes can reduce the risk of contamination. Sample processing time and cost can be reduced by using direct pathogen detection-without upstream RNA extraction by commercial kits. *Nie et al. (2012)* demonstrated that direct RT-LAMP assay can detect EV71 in heat-treated nasopharyngeal swab specimens.

Lastly, the sensitivity and specificity of the RT-LAMP assay can only be compared with RT-qPCR as there is no true gold standard for SARS-CoV-2 detection. It is possible that either test may have misclassified the true result of some of the samples. It is also important to note that the LAMP primers are highly specific, different primers may be needed for detection of different types of mutations of the target gene in the future.

SARS-CoV-2 can also be detected in other biological samples such as sputum, bronchoalveolar lavage fluid, feces and blood (*Wang et al., 2020*). Future studies are expected to assess the feasibility of RT-LAMP in detection of SARS-CoV-2 RNA extracted from these biological samples.

To conclude, an inexpensive, rapid, sensitive and specific RT-LAMP assay was successfully designed for SARS-CoV-2 detection. The simplicity of RT-LAMP combined with rapid turnaround time has shown it to be a valuable and applicable tool for the diagnosis of infectious diseases, particularly in resource-limited countries. In addition, RT-LAMP can be easily adapted to point-of-care diagnosis of COVID-19 as supported by a review written by *Nguyen, Bang & Wolff (2020)* and a non-peer-reviewed preprint by *Zhang et al. (2020)* using seven clinical samples. Early and accurate diagnosis is crucial to identify patients with COVID-19 for prompt institutionalization of infection control and public health measures, and when available, treatment.

## ACKNOWLEDGEMENTS

We thank the Director General of the Ministry of Health Malaysia for his permission to publish this article and Dr Xiao Teng Ching for technical support. The following reagent was obtained through the NIH Biodefense and Emerging Infections Research Resources Repository, NIAID, NIH: gamma-irradiated SARS-coronavirus, NR-9547.

### Funding

This study was supported by the Defense Threat Reduction Agency, USA under Broad Agency Announcement HDTRA1-6 (grant number HDTRA1-17-1-0027) and High Impact Research Grant (UM.C/625/1/HIR/MOHE/CHAN/14/3) from the Ministry of Higher Education Malaysia. The funders had no role in study design, data collection and analysis, decision to publish, or preparation of the manuscript.

## Grant Disclosures
The following grant information was disclosed by the authors:
Defense Threat Reduction Agency, USA (HDTRA1-6): HDTRA1-17-1-0027.
High Impact Research: UM.C/625/1/HIR/MOHE/CHAN/14/3.

## Competing Interests
The authors declare that they have no competing interests.

## Author Contributions
- Yee Ling Lau conceived and designed the experiments, performed the experiments, analyzed the data, prepared figures and/or tables, authored or reviewed drafts of the paper, and approved the final draft.
- Ilyiana Ismail performed the experiments, analyzed the data, authored or reviewed drafts of the paper, and approved the final draft.
- Nur Izati Mustapa performed the experiments, analyzed the data, authored or reviewed drafts of the paper, and approved the final draft.
- Meng Yee Lai performed the experiments, analyzed the data, authored or reviewed drafts of the paper, and approved the final draft.
- Tuan Suhaila Tuan Soh performed the experiments, analyzed the data, authored or reviewed drafts of the paper, and approved the final draft.
- Afifah Hassan performed the experiments, analyzed the data, authored or reviewed drafts of the paper, and approved the final draft.
- Kalaiarasu M Peariasamy conceived and designed the experiments, authored or reviewed drafts of the paper, and approved the final draft.
- Yee Leng Lee conceived and designed the experiments, authored or reviewed drafts of the paper, and approved the final draft.
- Yoong Min Chong performed the experiments, authored or reviewed drafts of the paper, and approved the final draft.
- I-Ching Sam performed the experiments, authored or reviewed drafts of the paper, and approved the final draft.
- Pik Pin Goh conceived and designed the experiments, authored or reviewed drafts of the paper, and approved the final draft.

## Ethics
The following information was supplied relating to ethical approvals (i.e., approving body and any reference numbers):

The Medical Research Ethics Committee (MREC) Ministry of Health Malaysia approved this research (NMRR-20-535-53855).

## Data Availability
Data are available in the Supplemental Files.

## Supplemental Information

Supplemental information for this article can be found online at http://dx.doi.org/10.7717/peerj.9278#supplemental-information.

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
