# Peer review of "Real-time reverse transcription loop-mediated isothermal amplification for rapid detection of SARS-CoV-2"

_PeerJ, doi:10.7717/peerj.9278_

## Round 0.1 · original submission · Major Revisions

Thank you for your important contribution to COVID19 science right now.

Please revise according to the reviewers' comments, especially the basic reporting requirement for FIGURES and DATA. Without figures, there is no evidence supporting the claims in this paper.

Also, please make sure to reference this paper: https://www.medrxiv.org/content/10.1101/2020.02.26.20028373v1.full

thank you! We will work hard to make this go fast!!

·

Basic reporting

This is a well-written submission. I have a few minor recommendations to improve the language in the article.

Line 72 - 73 - "the limit of detection (LODs) were" should be either singular or plural.

Line 73 - "previously published method" should be either singular or plural.

Line 77 - please define "LAMP-LF" before using an acronym.

Line 137 - Rather than "a cheap," I recommend "inexpensive."

Line 140 - Rather than "poor countries where resources are limited," I recommend "resource-limited countries."

Experimental design

Can you add information regarding how you obtained the nasopharyngeal swabs from patients? Did you use RNA extracted previously for RT-PCR and freeze the sample prior to testing, or did you obtain your own samples from COVID-tested patients? Etc.

Validity of the findings

Can you talk a little more about why you used the SARS-CoV2 N1 fragment as your choice for LAMP primer design? Typically researchers design LAMP primers to very highly conserved regions because LAMP primary cannot function if even a single SNP is present. Viruses mutate rapidly, how can you be sure that this region is highly conserved enough that future mutations will result in the primer set becoming non-functional? What evidence do you have that this region is highly conserved?

Can you expand the written component in your results section to include the names of the negative control viruses and the number of samples you tested so that the reader does not need to read through the full results section?

Additional comments

Thank you very much for contributing to the COVID-19 literature during this worldwide pandemic. In your report, please expand briefly on the down sides of the LAMP procedure compared to the gold standard for detecting microorganisms (RT-PCR). Specifically, it is important to note that the primers are HIGHLY specific and that any mutation can throw off the entire reaction. You also recommend the use of LAMP in under-developed countries, but can you discuss the limitations that some of these countries may have in completing this full protocol? How could those limitations be overcome? Finally, there is much recent evidence that SARS-CoV-2 RNA could be found in other biologic samples such as sputum, urine, throat swabs, and pulmonary lavage samples--a future option to study would be whether the LAMP reaction can take place in these types of samples.

·

Basic reporting

1. There are no figures. Please show the color change for the LAMP reaction (and preferably compare it to running on a gel) and some quantification of standards and samples from the RT-PCR reaction.

2. Please specify when the HNB was added to the RT-LAMP reactions for the experiments shown in the paper. The paper would be improved by showing that HNB added before amplification does not affect the reaction.

3. Please describe how RNA extraction was performed.

4. Please describe in detail how the in vitro transcribed RNA SARS-CoV-2 standards for RT-LAMP were made: primer sequences, conditions, etc.

Minor revisions:
Line 41: Specify U.S. CDC.
Line 53: Missing period
Line 72: limit should be limits
Line 97-98: “Addition of HNB to the LAMP reaction solution did not affect the time taken for amplification (data not shown).” Please move to the Discussion if not showing data.

Experimental design

Please see point 1. More description of the methods is required to replicate.

Validity of the findings

See point 1 - underlying data are not included.

Additional comments

This is an important contribution – isothermal protocols for COVID-19 diagnosis are critical. The high sensitivity and specificity of this RT-LAMP protocol are promising, and the use of HNB for assessment could be an important contribution. However, much more underlying data need to be shown and methods described so that this protocol can be fully assessed and replicated.

---

## Round 0.2 · Major Revisions

Please refer to the reviewer comments, especially with regard to including raw data.

·

Basic reporting

This revised article is clear and the points of clarification raised in the reviews were well-addressed. The results as described are so promising, and the development of a specific and sensitive RT-LAMP assay for SARS-CoV-2 that does not require opening the tube is really fantastic. However, sufficient raw data are not included. The following claims lack sufficient raw data:
1. The tubes of positive and negative samples shown in Figure 1 have a clear color difference, which makes this contribution very exciting. However, we need to see the tubes for all the assays reported in Supplementary Table 2 if those results are to be included throughout the paper (e.g., as part of the sensitivity and specificity calculation described in the abstract).
2. Similarly, all of the RT-PCR traces from Supp. Table 2 need to be shown. If possible, the quantitation from the trace should be shown as a bar graph or other readout in Fig.2 and the raw traces put in as a supplementary figure.
3. Still, technically, no raw data is shown for the claim that HNB does not inhibit the RT-LAMP reaction. However, now that the authors have clarified that all RT-LAMP reactions presented had HNB added at the beginning, this claim can just be cited as “data not shown.”
4. The raw data for the other coronaviruses tested is not shown.
5. The experiments used to determine the LOD for qRT-PCR and RT-LAMP are not shown.
Missing information from the methods: what primers and kits are used for qRT-PCR?
Missing information from the figures: please label the RT-LAMP primer locations in the clustal-w readout in Supp Fig 1.

Experimental design

The only point of clarification not yet addressed (I apologize that it wasn’t explicitly included in my earlier review) is how the samples were chosen for inclusion in the RT-LAMP experiments. For example, were positive samples with especially high titers by RT-PCR chosen? From looking at the current Figure 2, this doesn’t appear to be the case, which is fantastic, but a brief description of how the samples were chosen would put any concerns to rest.

Validity of the findings

See point 1. Much raw data is still missing.

Additional comments

This is an important contribution and PeerJ does not consider impact. If you no longer have any raw data I asked for, please consider how the statements of the paper can be changed so that they are supported by the data and the work can still be published.

---

## Round 0.3 · Minor Revisions

Thank you for your attention to the reviewer comments. I'd like to make sure that you clarify the goals of the study in the abstract. Please add a sentence in the methods describing the study, a description of clinical samples included and how many in each group, and the controls.

A minor comment: numbers less than 10 should be spelled out in the abstract and non-methods text. E.g line 30: 1 copy -> one (in the methods section 5 uL and other examples like that can be left alone)

---

## Round 0.4 · Minor Revisions

Thank you for carefully addressing the comments. One small issue to clarify: Please check that line 54 is correct (e.g is this the first instance of SARS-CoV-2 with RT-LAMP using HNB?). Perhaps better to say it is the first report to your knowledge?

---

## Round 0.5 · accepted · Accept

Congratulations! This is all moving very fast, and I appreciate your quick and responsive edits. Even more, I appreciate your efforts to develop much-needed testing methods. Good luck!